# Repeatome Analysis of Plasma Circulating DNA in Patients with Cardiovascular Disease: Variation with Cell-Free DNA Integrity/Length and Clinical Parameters

**DOI:** 10.3390/ijms26146657

**Published:** 2025-07-11

**Authors:** Stefania Fumarola, Monia Cecati, Francesca Marchegiani, Emanuele Francini, Rosanna Maniscalco, Jacopo Sabbatinelli, Massimiliano Gasparrini, Fabrizia Lattanzio, Fabiola Olivieri, Maurizio Cardelli

**Affiliations:** 1Advanced Technology Center for Aging Research, IRCCS INRCA, 60121 Ancona, Italy; s.fumarola@inrca.it (S.F.); r.maniscalco@inrca.it (R.M.); f.olivieri@staff.univpm.it (F.O.); 2Department of Human Sciences and Promotion of the Quality of Life, San Raffaele Roma Open University, 00166 Rome, Italy; monia.cecati@uniroma5.it; 3Clinic of Laboratory and Precision Medicine, IRCCS INRCA, 60121 Ancona, Italy; f.marchegiani2@inrca.it (F.M.); e.francini@inrca.it (E.F.); j.sabbatinelli@staff.univpm.it (J.S.); 4Department of Clinical and Molecular Sciences, Università Politecnica delle Marche, 60126 Ancona, Italy; 5Department of Agriculture, Food and Environmental Sciences, Polytechnic University of Marche, 60131 Ancona, Italy; 6Scientific Direction, IRCCS INRCA, 60124 Ancona, Italy; f.lattanzio@inrca.it

**Keywords:** cfDNA, Alu elements, repetitive DNA, cardiovascular disease, hearth failure, eGFR, CKD, circulating biomarkers, low-pass NGS

## Abstract

Repetitive DNA represents over 50% of the human genome and is an abundant component of circulating cell-free DNA (cfDNA). We previously showed that cfDNA levels and integrity can predict survival in elderly patients with cardiovascular disease. Here, we aimed to clarify whether a low-pass next-generation sequencing (NGS) approach can characterize the repeat content of cfDNA. Considering the bimodal distribution of cfDNA fragment lengths, we examined the occurrence of repetitive DNA subfamilies separately in dinucleosomal (>250 bp) and mononucleosomal (≤250 bp) cfDNA sequences from 24 patients admitted for heart failure. An increase in the relative abundance of Alu repetitive elements was observed in the longer fraction, while alpha satellites were enriched in the mononucleosomal fraction. The relative abundance of Alu, ALR, and L1HS DNA in the dinucleosomal fraction correlated with different prognostic biomarkers, and Alu DNA was negatively associated with the presence of chronic kidney disease comorbidity. These results, together with the observed inverse correlation between Alu DNA abundance and cfDNA integrity, suggest that the composition of plasma cfDNA could be determined by multiple mechanisms in different physio-pathological conditions. In conclusion, low-pass NGS is an inexpensive method to analyze the cfDNA repeat landscape and identify new cardiovascular disease biomarkers.

## 1. Introduction

More than half of the human genome is made up of highly repetitive DNA sequences. Once thought of as functionless, selfish DNA [1,2], repetitive sequences are now known to have important regulatory roles. In particular, “satellite DNA” repeats are dedicated to maintaining the integrity and function of centromeres and telomeres [3,4], while short and long interspersed repeat elements (SINEs and LINEs, respectively) enrich the genome with important regulatory sites and generate different classes of regulatory non-coding RNAs [5]. A loss of epigenetic control of repetitive DNA elements, observed in aging and cancer, exposes the risk of triggering somatic mutations and activation of inflammation by innate immunity mechanisms [6,7,8].

Repetitive sequences are also abundant in genome-derived circulating cell-free DNA (cfDNA), which is released into the blood by a variety of mechanisms. These include apoptosis, necrosis, neutrophil extracellular traps (NETs) formation, microvesicle and exosome release, and many others [9]. The specific mechanism of release of cfDNA in the blood is thought to affect the integrity and length of the cfDNA, which in most cases peaks between 160 and 170 bp (corresponding to the DNA contained in a chromatosome), but often contains smaller amounts of longer fragments [10,11]. Analysis of cell-free DNA is being considered as a tool for “liquid biopsy”, making it useful as a diagnostic or prognostic biomarker not only in cancer but also in age-related diseases [12,13,14]. Recent evidence suggests that repetitive DNA, particularly the satellite DNA component, is overrepresented in cfDNA and that its composition may change depending on specific disease conditions [15,16]. Because of their abundance, repetitive DNA elements have been targeted in sensitive methods to analyze cfDNA concentrations in blood. In particular, Alu elements, the most abundant interspersed SINE repeats with approximately one million copies for the human haploid genome [17] are targeted in a PCR-based method, aimed to detect the abundance and integrity of cfDNA [18] and widely used to obtain diagnostic or prognostic information in a variety of diseases [19,20,21,22,23,24,25,26]. The technique provides concentration parameters for two different Alu targets of different lengths (115 bp and 247 bp), with the ratio between the two providing a parameter of cfDNA integrity [18]. By this method we have previously shown that plasma cell-free Alu DNA concentration and integrity help to predict risk of death in patients hospitalized for acute cardiovascular events, in particular heart failure (HF) [26]. To clarify whether the increase in Alu cfDNA observed in high-risk patients also reflects a change in circulating cfDNA composition, and how cfDNA composition relates to cfDNA integrity in patients with heart failure, we decided to characterize the repetitive DNA content in a subset of samples selected from the same cohort. Using low-pass next-generation sequencing (NGS), we obtained a description of how the relative abundance of repetitive DNA families in plasma cfDNA can be variable between shorter and longer cfDNA fragments, and between HF patients with different clinical conditions, such as altered renal function parameters and presence/absence of chronic kidney disease (CKD) comorbidity. As this type of approach can be replicated in similar studies at relatively low cost, the present work may pave the way for the definition of increasingly specific and sensitive prognostic biomarkers for cardiovascular disease based on the analysis of circulating repetitive DNA sequences.

## 2. Results

### 2.1. Length Distribution of cfDNA

The length distribution of the cfDNA sequencing reads was bimodal for all the samples, and the modal lengths were around 170 and 320 bp (Figure 1A and Appendix A) with a minimum at 250 bp between the two peaks. Similar peaks were generally also visible in the high-sensitivity electrophoretic analysis of the same cfDNA samples and were assumed to correspond, based on previous literature evidence [27], to mononucleosomal (≤250 bp) and dinucleosomal (>250 bp) cfDNA fractions (Figure 1B and Appendix A), respectively.

### 2.2. Overall Abundance and Distribution of Repetitive Elements in Cell-Free DNA

The repetitive DNA composition of cfDNA was analyzed by aligning 100 bp sequences extracted from sequencing reads to a list of consensus sequences of repetitive DNA families in the human genome obtained from the RepBase database. Overall, 23.20% of the cell-free DNA-derived sequences mapped to a repeat element family/subfamily consensus sequence (Figure 2A). This percentage was approximately half of the 47.13% of cfDNA-derived sequences that mapped to repetitive element annotations on the Genome Reference Consortium Human Build 38 (GRCh38) (Figure 2B). For comparison, we calculated that the percentage of repetitive elements in GRCh38 is 54.40% (Appendix A).

The overall repeat composition in cfDNA sequences is shown in Figure 3A. The repeat composition of the reference genome is also shown for comparison (Figure 3B). Among the cfDNA sequences mapped to the consensus list of repetitive DNAs, sequences derived from LINE-1 and Alu families were the most abundant, with 30.04% and 23.56%, respectively (Figure 3A). These values were similar to the percentages of these two repetitive families (33.95% and 20.42%, respectively) in the repetitive part of the human genome reference sequence (Figure 3B). Conversely, satellite DNA families showed a much higher frequency in the repetitive fraction of cfDNA (17.35%) than in the repetitive fraction of the reference genome (4.89%). The THE1 (3.73% vs. 1.73%) and HERV (2.93% vs. 1.48%) subfamilies showed a lower percentage increase in the repetitive cfDNA fraction compared to the repetitive genome fraction, whereas the MIR (0.45% vs. 5.41%), LINE-2 (0.19% vs. 6.67%), and MLT (2.90% vs. 4.88%) sequences were present in a lower percentage in the repetitive cfDNA compared to the repetitive genome. The variation in the percentages of the other categories (MST, LTR-ERV, MER, rRNA, etc.) between repetitive cfDNA and repetitive genome was equal to or less than 1%.

### 2.3. Variable Distribution of Repetitive Elements in Short and Long cfDNA Fragments and Among Different Samples

Further analyses were conducted to compare the repetitive composition of different cfDNA fractions (mono- vs. dinucleosomal fragments) in different samples. For these analyses a “RepBase reduced list”, containing the consensus of only the most abundant repeat subfamilies in the cfDNA (Appendix A), was used as a mapping target. The cfDNA (100 bp sequence fragments) mapping to the reduced list was composed predominantly by Alu and other SINEs (43.53%) and by satellite DNA (44.35%), while the other major repeat families were present, but less represented (Appendix A). The percentage of cfDNA sequence mapping to each subfamily, within a repeat family, is shown in Appendix A.

Analysis of the composition of repetitive sequences in mono- and dinucleosomal cfDNA fragments of all samples (Figure 4) shows that most of the major repetitive DNA classes or “family groups” (defined as in the corresponding column in Appendix A) had a different distribution between short and long cfDNA fragments.

Figure 4 and Appendix A, show that Alu subfamilies (here classified as Alu-J, Alu-S, and Alu-Y) were enriched in longer cfDNA fragments compared to shorter fragments, and the same bias in distribution was observed for SVA SINEs, for ribosomal DNA, for HERV-associated LTRs and other ERV-like repeats, and for DNA satellites found in pericentromeric regions or acrocentric chromosomes. In contrast, centromeric satellites (ALR) and L1 subfamilies, with the exception of L1HS, were increased in short compared to long cfDNA fragments. Variations observed in HERV, L1HS, and HSAT were not significant after Bonferroni correction.

### 2.4. The Relative Abundance of Alu cfDNA Is Inversely Related to cfDNA Integrity

When analyzing the correlation between the relative abundance of repetitive cfDNA derived from the Alu elements and different parameters of cfDNA abundance and integrity (Table 1), we found that Alu DNA tends to decrease its relative abundance in both short and long fragments of cfDNA as the integrity of the cfDNA increases. This finding holds regardless of the method used to assess cfDNA integrity (real-time PCR 247/115 ratio, ratio of electrophoretic peaks of cfDNA analyzed by Agilent Tape Station, or ratio between long and short reads in the cfDNA library). When considering three major Alu subfamilies (Alu-J, Alu-S, and Alu-Y) instead of the whole Alu family, the correlation of the relative abundance with integrity is always inverse (Spearman rho < 0), but the statistical significance of this correlation is reached for all the cfDNA integrity measurement methods only when considering the Alu-S subfamily, while it is method-dependent for other Alu subfamilies (Table 1). In most cases, the inverse correlation between cfDNA integrity and relative Alu abundance is stronger when the Tape Station method is used to analyze cfDNA integrity (Table 1). The integrity calculated by this method is also inversely correlated (Table 1) with the fold change of the CPM of the Alu family and all Alu subfamilies in long vs. short cfDNA. In addition, the total cfDNA concentration measured by Alu115 real-time PCR (Table 1) correlates positively with the relative abundance of dinucleosomal Alu-J and with the fold change of Alu (total) and Alu-J sequences in the dinucleosomal vs. mononucleosomal fraction.

### 2.5. The Composition of cfDNA Can Vary in Correlation with Clinical Biomarkers

The variation in Alu cfDNA composition was then analyzed in relation to the clinical conditions of the patients. The clinical conditions and laboratory characteristics of the patients included in the study, as well as a stratification of the main clinical and biochemical parameters with respect to outcome, are shown in Appendix A, respectively.

When the cfDNA repeat distribution is analyzed in relation to parameters of clinical risk it is possible to observe a direct correlation between CPM of Alu elements (total Alus, Alu-S, and Alu-Y subfamilies) in long cfDNA and eGFR (Table 2 and Appendix A), and an inverse correlation between CPM of Alu elements (total and all the three subfamilies) in long cfDNA with creatinine (Table 2). In particular, the most significant results (*p*-value < 0.001) were observed for total Alus and Alu-S. For short cfDNA, similar correlations with eGFR and creatinine are found only for the Alu-Y subfamily, but with a lower significance value (*p* < 0.05). In addition, the same parameters of renal function showed a correlation with the integrity of cfDNA measured by Tape Station analysis (the ratio between the dinucleosomal and the mononucleosomal peaks) (Appendix A). Concerning the other repeat families, highly significant (*p*-value < 0.001) correlations were found in dinucleosomal cfDNA for L1HS with troponin and for ALR with Nt-proBNP. Other significant correlations, albeit at a lower significance level (*p* < 0.05) were found for ALR (in both mononucleosomal and dinucleosomal fractions) with NLR, and for the “other ERV-like” category with CRP (only in the mononucleosomal fraction).

As the most significant correlations with prognostic biomarkers were found to involve Alu, L1HS and ALR cfDNA parameters, we decided to evaluate the association of these parameters with survival. Additionally, given the observed correlation between renal function parameters (e-GFR, creatinine) and Alu cfDNA composition, we decided to evaluate the Alu composition of cfDNA from patient subgroups characterized by different renal comorbidity status. While the differences between patients with different survival did not reach significance for any of the tested cfDNA parameters (Appendix A), the CPM of Alu DNA was lower in patients with CKD than in those without this comorbidity, for both short and in long cfDNA (Table 3).

When analyzing the three main Alu subfamilies instead of total Alus, the variation in Alu-S retained its significance after Bonferroni adjustments in both long and short cfDNA, whereas Alu-J was significantly different between the two groups only in short cfDNA. In addition, a decrease in total Alu and Alu-S relative abundance was observed in dinucleosomal with respect to mononucleosomal cfDNA fragments in patients with CKD comorbidity (Appendix A).

## 3. Discussion

We found evidence that in patients with cardiovascular disease, the dinucleosomal fraction of circulating cfDNA has a different repetitive DNA content compared to the mononucleosomal fraction, with an increase in Alu elements and a decrease in centromeric satellites and LINE-1 elements. It is well known that in the human genome, Alu SINEs are preferentially found in gene-dense regions, whereas LINE-1 elements are enriched in gene-poor regions [17], so our result fits with recent data showing that long cfDNA molecules (longer than 500 bp) preferentially originate from transcriptionally active regions of the genome [28]. Considering that another study reached different conclusions when analyzing the variation of composition with length within the mononucleosomal DNA [29], we could speculate that the relationship between repetitive composition and cfDNA length may differ between different cfDNA length ranges.

Regarding the relationship between cfDNA repeat composition and clinical parameters, we found a positive correlation of Alu relative abundance with eGFR (mainly in the dinucleosomal cfDNA fraction) and an association of reduced Alu relative abundance with CKD comorbidity (in both cfDNA fractions). Such results suggest that, in patients with HF, impaired renal function is associated with reduced percentage of Alu in repetitive cfDNA. Given that renal dysfunction is an important prognostic marker for adverse outcomes in patients with HF [30], these observations suggest investigating whether Alu relative abundance in repetitive cfDNA can in turn constitute a new prognostic factor in these patients. In addition, increasing cfDNA integrity (especially when measured by high-sensitivity electrophoresis as increasing relative abundance of dinucleosomal cfDNA compared to mononucleosomal cfDNA) was associated with decreased relative abundance of Alu cfDNA, mainly in the dinucleosomal fraction, and with decreased renal function as measured by eGFR. Thus, renal dysfunction in this cohort of patients with HF appears to be associated with an increased abundance of Alu-depleted dinucleosomal cfDNA. As different mechanisms of generating plasma cfDNA may contribute to its repetitive element composition [15], this finding could be interpreted as an increased entry of DNA fragments from Alu-poor genomic regions into the circulation, or as reduced protection from the degradation of Alu-rich genomic regions. Genomic Alu elements are typically occupied by one or two nucleosomes [31], which can offer partial protection to the corresponding cfDNA against nuclease cleavage [10]. The nucleosome occupancy of Alu elements can decrease due to specific cellular mechanisms, such as the binding of the transcription factor TFIIIC [32]. Therefore, physiological or pathological conditions that affect the binding of TFIIIC to Alu-rich regions could potentially impact the relative abundance of these regions in cfDNA, particularly in longer fragments. Alternatively, an alteration in the renal clearance mechanisms of cfDNA could be hypothesized. Although the clearance of cfDNA is thought to be only partly based on renal clearance, plasma cfDNA may be trapped in the glomeruli by a size-selective mechanism of unknown nature [33,34], so it is possible that renal dysfunction may selectively alter cfDNA clearance based on its length or composition.

Concerning the specific results obtained on the main Alu subfamilies, the most significant results regarding the correlation of relative repeat abundance with cfDNA integrity, renal function parameters, or the presence of CKD comorbidity involve the Alu-S subfamily more often than the Alu-Y and Alu-J subfamilies. In addition to their different evolutionary ages, Alu-S, Alu-J, and Alu-Y exhibit distinct genome distributions and chromatin-binding properties. These include different enrichment levels in open chromatin and gene-rich regions and at the boundaries of topologically associated domains (higher for Alu-S and Alu-J), different affinities for H1.4/H1X histone binding (higher for Alu-Y and Alu-S), and variable densities across different chromosomes [35,36,37,38,39,40,41]. Moreover, as nucleosome binding to Alu sequences appears to be sequence-dependent [31], it is reasonable to suppose that the greater conservation of Alu-S and Alu-Y sequences, compared with the older, more frequently mutated Alu-J elements, could result in higher potential for nucleosome binding and lower susceptibility to cfDNA degradation. Overall, it would be interesting to investigate whether this particular combination of genomic distribution and chromatin-binding properties of the Alu-S subfamily renders it more susceptible to changes in cf-(dinucleosomal)-DNA abundance in specific disease conditions, compared to the other Alu subfamilies.

In addition to the observed association between Alu cfDNA and biochemical parameters of renal function, we found that the relative abundance of L1HS and ALR repeat families in dinucleosomal cfDNA was strongly correlated with two other prognostic biomarkers: troponin and NT-proBNP, respectively. This suggests that repeat families with different genomic distributions could provide cfDNA-related biomarkers with different biological meanings and potential uses in cardiovascular patients. However, the lack of direct association between these cfDNA parameters and outcome requires clarification. The presence of other factors, such as comorbidities or inflammation, could potentially interfere with the relationship between cfDNA composition and prognostic biochemical or clinical parameters. However, none of the considered cfDNA parameters (Alu, ALR and L1HS relative abundance in dinucleosomal cfDNA) correlated with other comorbidities, while only ALR relative abundance showed a weakly significant correlation with an inflammatory parameter (NLR). Nevertheless, larger-scale studies are required to determine whether inflammation or other comorbidities (other than CKD) contribute to these findings.

It is worth noting that the percentage of cfDNA repetitive sequences obtained in this study was similar (47.13% vs. 52.5%) to the percentage of the human genome reference classified as repetitive by RepeatMasker annotations [42]. Other studies have reported a higher proportion of repetitive elements in cfDNA. For instance, Grabuschnig et al. [15] found a repetitive fraction of 78.49% in cfDNA obtained from healthy individuals. This difference is probably due to methodological differences in data processing and annotation.

In the analysis of overall repeat composition of cfDNA, the most abundant repeat families (Alu and LINE-1 elements) showed a very similar relative abundance (with a variation of 3–4%) compared to their representation in the genome reference [42]. However, the relative abundance of DNA satellites in our cfDNA samples was more than three times higher than that reported for the human genome. This observation is consistent with the results of other studies [15,43,44,45]. This enrichment of satellite DNA relative to the reference genome (GRCh38) was suggested to be the result of a specific mechanism acting on centromeric DNA during DNA replication, involving replication fork stalling at centromeric satellite sequences, formation of secondary structures, and clearance of these structures via exosomes [15,46]. In addition, at least some of the satellite DNA subfamilies tend to be underrepresented in genomic references due to the difficulty of correctly incorporating tandem repeats into genomic assemblies [47], which may lead to an apparent overrepresentation in cfDNA.

We also found a lower representation of some repeat families, in particular MIR and LINE-2, in our cfDNA samples compared to their prevalence in the reference genome. This could be due to the method adopted for analyzing cfDNA repeat composition, which is based on mapping against repeat family consensus sequences. The old evolutionary age of the MIR and LINE-2 repeat families [48,49] implies low conservation of the elements in relation to their family consensus sequence, and a possible lower mapping efficiency. Indeed, in the human genome, the divergence from the consensus is greater than 37% for MIR elements and greater than 46% for LINE-2 elements [50]. For comparison, the divergence of Alu-Y elements from their consensus is only around 4.3% [50].

The demonstration that low-pass NGS analysis of plasma cfDNA allows characterization of the repetitive landscape of cfDNA samples can be considered a methodological achievement. In our approach, the high copy number of targets (repetitive elements) compensated for the low genomic coverage of the NGS reads, allowing the detection of significant differences between samples or sample fractions. The analytical approach adopted (i.e., mapping cfDNA sequences to lists of repetitive subfamily consensuses rather than to the annotated genome reference) allowed the analysis of about half of the reads corresponding to repetitive genomic elements. Although most studies on this topic have used a different approach, i.e., NGS at high sequencing depth followed by mapping of sequencing reads (or sequence-derived K-mers) to the genome reference annotated with repetitive element positions [15,16], at least one other publication prior to this one has explored the possibility of mapping repetitive cfDNA to a list of consensus subfamilies [30]. To compare the composition of mononucleosomal and dinucleosomal cfDNA without different read lengths affecting mapping to different repeat family/subfamily consensus sequences, we implemented an “in silico” strategy of uniformly fragmenting cfDNA sequencing reads into 100 bp segments. This approach effectively normalizes length differences between repeat families, allowing mapping results to reflect sequence homology with consensus sequences rather than fragment size.

To our knowledge, this is the first study to apply this type of approach in a clinical context, paving the way for a relatively low-cost NGS analysis of the repetitive cfDNA landscape in clinical settings. It should be noted that the small sample size analyzed is motivated by the exploratory purpose of this study, i.e., to assess the feasibility of low-pass NGS sequencing to analyze the cfDNA repetitive landscape in patients with HF. Any associations between clinical data and cfDNA parameters presented here should be considered in this context. For instance, when the estimated effect size of the difference in total Alu CPM between patients with and without CKD is calculated, the result (taking into account the 95% confidence intervals of this parameter) is a variation of almost 20-fold. Hence, the observed associations between the cfDNA repeatome and clinical data should be treated with caution and require confirmation in larger studies.

## 4. Materials and Methods

### 4.1. Study Population

The patient data and biological samples included in this study have been collected as part of the Report-Age project (trial registration number NCT01397682), an observational study on the health status of older adult patients (aged 65 years and older) admitted at the National Centre for Aging (IRCCS INRCA) hospital, Ancona, Italy, initiated in September 2011 [51]. The study was conducted according to the guidelines of the Declaration of Helsinki. For this analysis, a subset of 24 patients was extracted from the larger cohort of 244 geriatric patients with cardiovascular disease included in our previous study [26]. Diagnoses at admission and comorbidities were coded according to the ICD, 9th revision [52]. Specifically, the patients in the extracted subgroup were admitted to the INRCA hospital for heart failure (ICD9 diagnosis codes 4281, 4289, 40201, 40211, 40291, 42830, 42841) between 2012 and 2017. Biological (plasma) samples were taken within the first 24 h of admission, and clinical data was collected, including 24-month follow-up mortality data. The extracted subgroup of 24 subjects was selected to include an equal number of patients with different outcomes (12 patients survived the entire follow-up period, 12 patients died before the end of follow-up), balanced by sex and in the same age range (>80 years). Among the biochemical parameters, eGFR was calculated using the Berlin Initiative Study 1 (BIS-1) equation, which is optimized for calculating eGFR in older adults, particularly those older than 70 years) [53].

### 4.2. Extraction and Quality Control of cfDNA

Plasma separation from EDTA blood samples and cfDNA extraction from plasma samples were performed as described in a previous publication [26] and extracted cfDNA samples were stored at −80 °C. A TapeStation System 4200 automated electrophoresis platform and Cell-free DNA ScreenTapes (Agilent, Santa Clara, CA, USA) were used to quantify the extracted cfDNA samples and analyze their quality in order to exclude possible contamination by high-molecular-weight (HMW) DNA that could interfere with the following steps.

### 4.3. cfDNA Quantification and Integrity Assessment

Real-time PCR analysis of Alu target sequences of 115 bp and 247 bp was used to obtain two parameters of plasma cfDNA concentration for total and long Alu cfDNA (“ALU 115” and “Alu 247” concentrations, respectively) and one parameter of cfDNA integrity (“Alu 247/115”). The analysis was performed as previously described [26].

In addition, the integrity of the cfDNA was also determined using two other methods:(1)Samples were analyzed by automated electrophoresis using high-sensitivity D1000 screen tapes on the TapeStation System 4200 (Agilent, Santa Clara, CA, USA); integrity was determined by the ratio of the molar concentration of the longer cfDNA peaks (size range 280–700 bp) to that of the shorter main peak at approximately 170 bp (size range 100–280 bp).
$$Integrity\left(by\;screen\;tape\right)=\frac{concentration[280-700]}{concentration[100-280]}$$(2)After NGS sequencing, integrity was also calculated as the ratio of the counts of reads in two size ranges (reads > 250 bases; reads ≤ 250 bases).$$Integrity\left(by\;reads\;count\right)=\frac{countofreads[>250bases]}{countofreads\left[\leq 250bases\right]}$$


### 4.4. Library Preparation and Next Generation Sequencing

Library preparation was performed using the QIAseq cfDNA Library T Kit (Qiagen, Hilden, Germany) starting from 0.4 to 1.0 ng of cfDNA. In particular, end-prep, ligation, library purification and library amplification were performed according to the manufacturer’s protocol. After amplification, libraries were purified using 1:1 MagSi-NGS Prep Plus beads (Magtivio, Nuth, The Netherlands) with 3 washes in 500 μL EtOH 70% and elution in 17 μL Qiagen Buffer EB (Tris HCL, PH 8). A second round of purification was performed (after dilution in IDTE to a total volume of 50 μL) with 1.2 × MagSi-NGS Prep Plus beads, 3 washes in 500 μL EtOH 70% and elution in 10 μL IDTE buffer (10 mM Tris, 0.1 mM EDTA). After checking the quality and concentration of each library using D1000 screen tapes on a TapeStation System 4200 (Agilent, Santa Clara, CA, USA), the libraries were pooled at a concentration of 100 pmol/L and finally diluted to 70 pmol/L for sequencing. Clonal amplification and loading onto an Ion 530 chip (Thermo Fisher Scientific Inc., Waltham, MA, USA), and NGS sequencing, were performed on an Ion Torrent Ion Chef and a ION S5 instrument, respectively (Thermo Fisher Scientific Inc., USA). A Chef Protocol—400 bp and 850 sequencing flows were used to obtain the maximum reads length possible on the instrument. A good quality run was ensured, with 82% ISP loading density and approximately 11 million reads. Approximately 290,000 to 800,000 reads were obtained for each cfDNA library.

### 4.5. Bioinformatic Analyses of NGS Data

After importing fastq files from the Ion S5 instrument server, NGS data analysis was performed using CLC Genomics Workbench version 7.0 software (Qiagen, Hilden, Germany). An initial quality trim was performed using default parameters. The sequencing reads were then used in different analysis workflows on the same software, as described and summarized below (Figure 5).

### 4.6. Quantification of Repetitive Elements in cfDNA and in the Human Reference Genome

To determine the percentage of repetitive DNA elements in cfDNA sequences, the library reads of all samples were mapped as a single pool against the GRCh38 human reference genome, using the CLC Workbench mapping tool “map read to reference” with the length fraction and similarity fraction parameters set to 0.9 and 0.9, respectively. Subsequently, repetitive DNA elements were obtained using the “Extract reads based on overlap” CLC tool. The “Overlap” parameter was set to “Within region” in order to select reads that completely overlapped the RepeatMasker annotation (version hg38—December 2013—RepeatMasker open-4.0.5—Repeat Library 20140131) downloaded from the RepeatMasker web site [42]. The percentage of non-repetitive DNA elements was calculated using the same tool, but with the “Overlap” parameter set to “No overlap”.

The distribution of repetitive elements in the genome GRCh38 was calculated (to be compared with that observed in cfDNA) by using the RepeatMasker annotations concerning the length and the type of each repeat element. In particular, the lengths of all individual repetitive elements (or the length of all individual repetitive elements of a given family) annotated on the genome were summed to have the total length of repetitive genome sequence (or the total length of repetitive sequence of a given family).

### 4.7. Analysis of Repeat Family/Subfamily Composition in cfDNA

The analyses of repeat family/subfamily composition in cfDNA were conducted by mapping cfDNA sequences to lists of repetitive family/subfamily consensus sequences, as detailed below.

In order to obtain mapping results unaffected by the different lengths of cfDNA fragments and cfDNA library reads, we preliminarily extracted 100 bp fragments from each sequencing read through an automated workflow (Appendix A). Reads with a length < 100 bp were discarded. The collection of 100 bp fragment reads obtained from all the samples, or from each single sample, was used for the following mapping analyses.

In particular, the distribution of cfDNA sequences in repetitive DNA families was initially analyzed by mapping 100 bp sequences extracted from all the cfDNA libraries to a list of 1355 human-specific and human-ancestral repeat family consensus sequences. The list was downloaded from the RepBase Update database of repetitive elements [54], a service of Genetic Information Research Institute (GIRI). The analysis was performed using the “map reads to reference” tool with the following parameters: length fraction and similarity fraction set to 0.3 and 0.9, respectively.

To obtain the percentage of sequences mapped to a particular repeat family (or subfamily group), the number of sequences mapped to (consensus of) member subfamilies was summed, and this sum was divided by the total number of sequences mapped to the whole list of consensus sequences. More precisely, the percentage “x*_i_*” of sequences mapped to a particular repeat family group “i” was calculated as follows:$$x_{i}=\frac{\sum_{n=1}^{N_{i}}\left(a_{n}\right)}{\sum_{i=1}^{k}\sum_{n=1}^{N_{i}}\left(a_{n}\right)}*100\%$$
where “a*_n_*” is the count of reads mapping to an “n” subfamily of the family (or subfamily group), “N*_i_*” is the number of subfamilies contained in the family (or subfamily group), and “k” is the number of families (or subfamily groups) in which the repeat subfamilies are classified.

To calculate, for comparison, the percentage of the repetitive human genome sequence annotated with repeat elements of the families (or subfamily groups) previously considered, the sum of the bp length of each repeat element of each family (or subfamily group) annotated on the reference genome with RepeatMasker was calculated, and then divided for the total length (in bp) of all the repetitive elements annotated on the reference genome. The used equation was:$$p_{i}=\frac{\sum_{n=1}^{S_{i}}\left(L_{n}\right)}{\sum_{i=1}^{k}\sum_{n=1}^{S_{i}}\left(L_{n}\right)}*100\%$$
where “L*_n_*” is the length in bp of an element of the family (or subfamily group), “S*_i_*” is the number of elements of that family (or subfamily group) in the genome, and “k” is the number of families (or subfamily groups) in which all the repeat elements annotated on the genome have been classified.

To allow comparison of cfDNA repeat variation between different samples or different sample fractions (long vs. short sequences), the RepBase list of 1355 human repeat families was reduced to obtain a “RepBase reduced list”, containing only the most abundant repeat subfamilies in the cfDNA population. For this purpose, a de novo assembly approach was applied to the entire set of cfDNA sequences [45], in order to identify the most abundant repetitive subfamilies in cfDNA. Standard parameters with automatic words and bubble sizes were used for the de novo assembly. Contigs with over 500 reads were selected, and their consensus sequences aligned to the complete list of 1355 human-specific and human-ancestral repeat family consensus sequences, obtained from RepBase Update. The mapping parameters were the same as reported above. The RepBase subfamilies with at least 1 contig mapping to their consensus were included in the “RepBase reduced list” used for the following mapping analysis.

Considering the bimodal distribution of cfDNA fragment lengths, we divided the cfDNA sequences into mononucleosomal cfDNA (≤250 bp) and dinucleosomal cfDNA (>250 bp) and separately analyzed the two fractions after extracting 100 bp sequence fragments from each read (Appendix A). The mapping of the cfDNA 100 bp fragments was performed using the RNA-SEQ analysis CLC tool, using the “RepBase reduced list” as the reference sequence; the parameters for the analysis included: length fraction and similarity fraction set to 0.3 and 0.9, respectively.

### 4.8. Statistical Analyses

The difference in relative abundance (assessed as CPM, count per million) of repetitive subfamilies in mononucleosomal (≤250 bp) and dinucleosomal (>250 bp) cfDNA fractions was tested using the paired sample *t*-test. A Bonferroni adjustment was applied to correct for multiple testing. Spearman’s correlation was calculated to measure the strength and direction of monotonic association between two variables. The Mann–Whitney U test was used to test for differences in the distribution of the counts per million (CPM) between sample groups (different for survival or comorbidity status). Again, the Bonferroni adjustment was used to correct for multiple testing.

The distribution analysis of family groups between short and long cfDNA fragments was conducted with heatmap analyses using the website www.heatmapper.ca. The Z score is calculated as number per million (CPM); the average linkage and Spearman Rank correlation were used for representation.

All the other statistical analyses were conducted with the SPSS version 29 software (IBM, Armonk, NY, USA).

## 5. Conclusions

The present results show that the low-pass NGS sequencing of cfDNA makes it possible to analyze the repetitive landscape of cfDNA. The relatively low cost and simplicity of this approach paves the way for its application in the search for new circulating biomarkers in a variety of conditions and even in large-scale studies. In particular, the finding that the relative abundance of Alu varies with cfDNA length improves the characterization of this circulating biomarker. In addition, the observed association of Alu cfDNA relative abundance with parameters of renal failure suggests a possible diagnostic/prognostic potential of this parameter. Further studies in larger cohorts should be performed to confirm these findings and to clarify whether they are restricted to patients with HF or could have a more general validity in older adults.

## Figures and Tables

**Figure 1 ijms-26-06657-f001:**
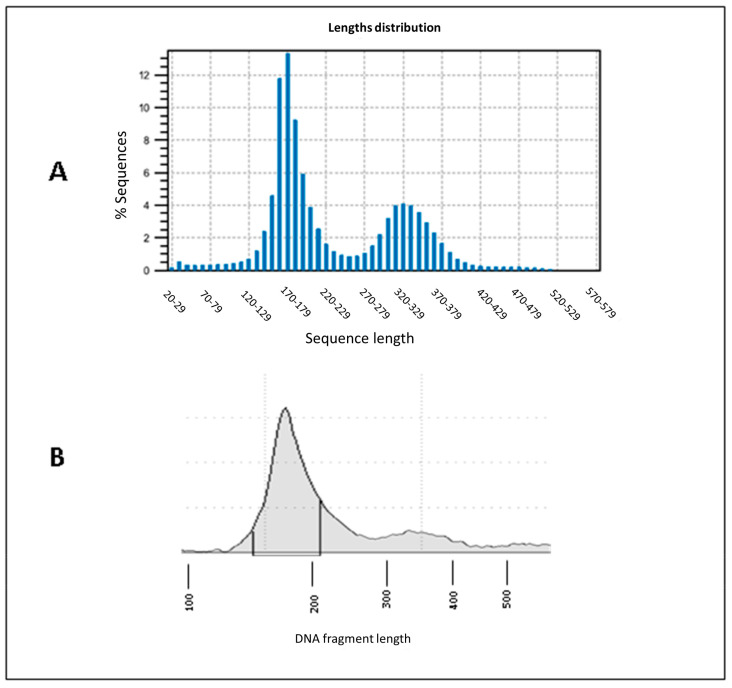
Graphical representation of the bimodal distribution of cfDNA. (**A**) Length distribution of NGS library reads of an example cfDNA (sample 2). (**B**) Automated electrophoresis of the same cfDNA sample conducted by high-sensitivity D1000 screen tapes on TapeStation System 4200. The two peaks represent the abundance of mononucleosomal (≤250 bp) and dinucleosomal (>250 bp) cfDNA fragments. Graphical representation of the length distribution of the other cfDNA samples and NGS libraries are shown in Appendix A.

**Figure 2 ijms-26-06657-f002:**
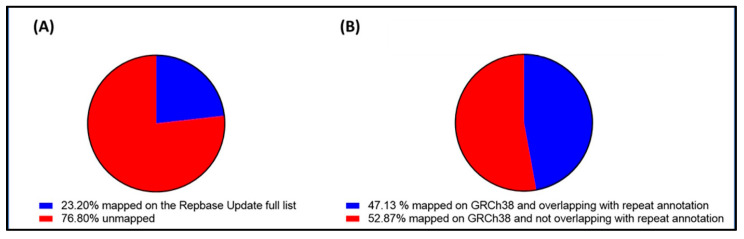
Representation of repetitive and non-repetitive elements in cell-free DNA and in GRCh38. The pie charts illustrate the overall percentage distribution of cfDNA sequencing reads mapped to the full list of 1355 human and ancestral repetitive subfamilies by RepBase Update (**A**) or mapped to the human reference genome (GRCh38) and overlapping (100% overlap of mapped sequence) with a RepeatMasker annotation (**B**).

**Figure 3 ijms-26-06657-f003:**
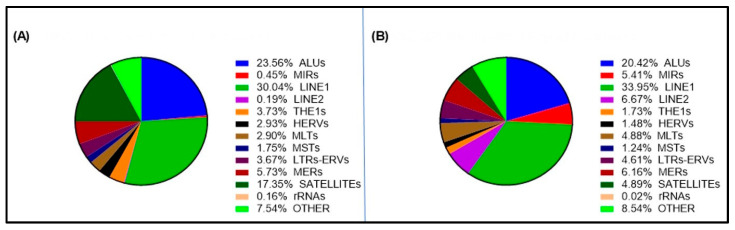
Representation of repetitive element subfamilies in cell-free DNA and in GRCh38. (**A**) The pie chart illustrates the overall distribution of cfDNA 100 bp fragment sequences mapped to specific DNA repeat categories (families or groups of repeat subfamilies). The total (100%) represents all the sequences mapped to the whole list of 1355 human-specific and human-ancestral consensus sequences obtained by the RepBase Update database. For the same repeat categories/subfamilies, the pie chart in (**B**) shows the percentage of sequence of the human genome reference annotated with repeat elements of the considered category/subfamily. In this case, 100% represents the total of the genome reference sequence that has been annotated with repetitive DNA elements.

**Figure 4 ijms-26-06657-f004:**
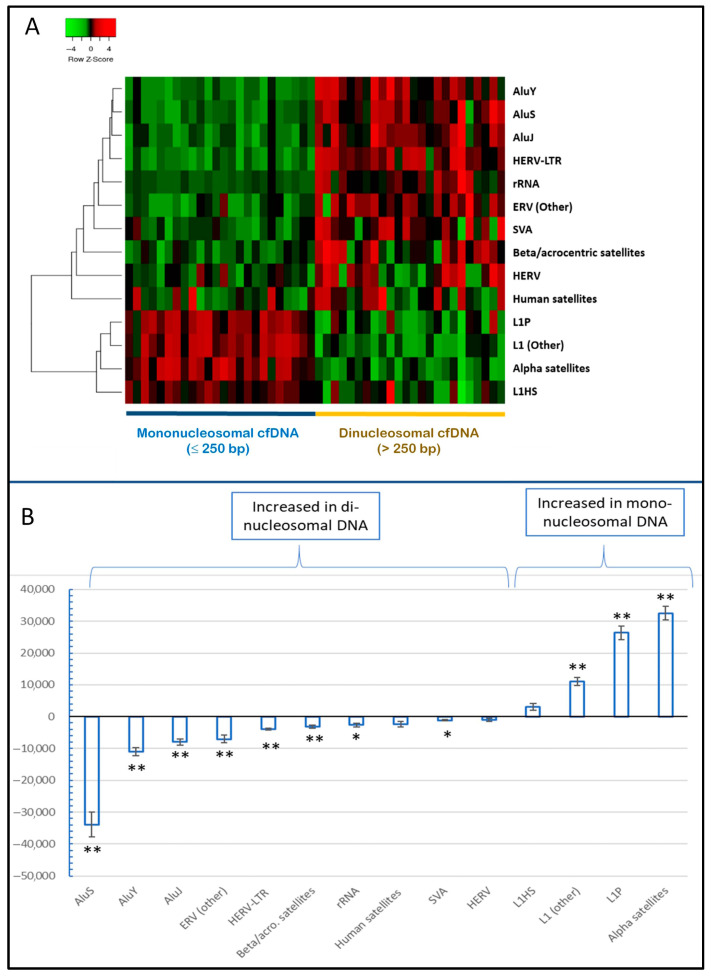
(**A**) Differential distribution of major repetitive DNA classes between mononucleosomal and dinucleosomal cfDNA fragments. The heatmap was obtained by mapping reads to the RepBase reduced list. The Z score is calculated as Count per million (CPM). Average linkage and Spearman Rank correlation were used for representation. (**B**) Mean difference (±S.E. of mean) in CPM of major repetitive DNA classes between mononucleosomal and dinucleosomal cfDNA sequences. Asterisks indicate differences significantly different from 0 (paired sample *t*-test), after Bonferroni correction: * = *p* < 0.01; ** = *p* < 0.001.

**Figure 5 ijms-26-06657-f005:**
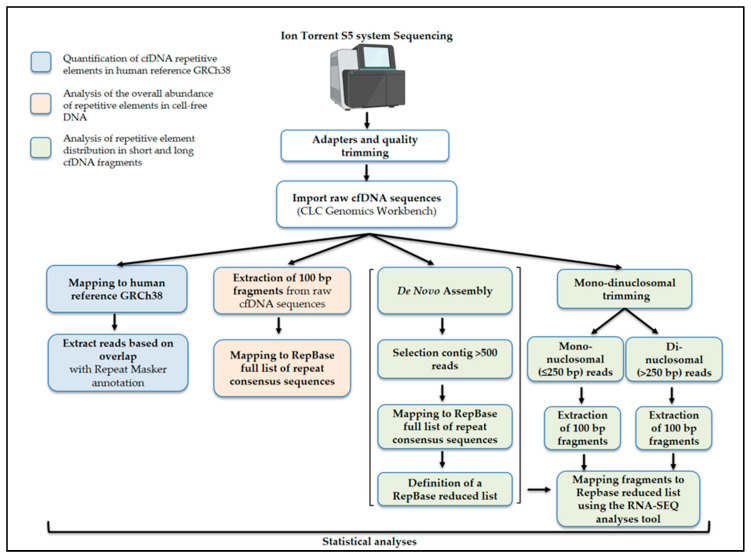
Bioinformatics pipeline. The flowcharts summarize the bioinformatic analyses performed using CLC Genomics Workbench software, version 7. The analyses were designed to (i) quantify repetitive elements in cfDNA by counting the number of reads mapped against the reference genome annotated by RepeatMasker (shown in blue); (ii) characterize the composition of repeat families and subfamilies in cfDNA by aligning reads to RepBase full list of repeat consensus sequences (orange); and (iii) assess the distribution of repetitive elements across short and long cfDNA fragments. The latter was achieved by first performing a de novo assembly to generate a reduced RepBase dataset, followed by separate mapping of mono- and dinucleosomal reads against these custom reference sequences (green).

**Table 1 ijms-26-06657-t001:** Correlation (Spearman rho) between relative abundance (CPM) of Alu families in cfDNA and different cfDNA parameters of abundance and integrity.

		Real-Time PCR Alu 115 (Plasma Concentration of Total Alu cfDNA)	Real-Time PCR Alu 247 (Plasma Concentration of Long Alu cfDNA)	Real-Time PCR Alu 247/115 (Alu cfDNA Integrity)	cfDNA_Integrity by H.S. Screen Tape ^a^	cfDNA_Integrity by NGS Reads Length
**CPM in mononucleosomal cfDNA**	**Alu-J**	0.168	0.020	−0.296	−0.400	**−0.479 ***
	**Alu-S**	0.189	−0.039	**−0.555 ****	**−0.497 ***	**−0.615 ****
	**Alu-Y**	0.001	−0.092	−0.100	**−0.453 ***	−0.325
	**Alu (total)**	0.145	−0.043	**−0.438 ***	**−0.522 ***	**−0.571 ****
**CPM in dinucleosomal cfDNA**	**Alu-J**	**0.499 ***	0.332	−0.313	**−0.571 ****	−0.384
	**Alu-S**	0.257	0.053	**−0.408 ***	**−0.675 ****	**−0.590 ****
	**Alu-Y**	0.356	0.151	−0.348	**−0.792 ****	−0.352
	**Alu (total)**	0.398	0.174	**−0.434 ***	**−0.765 ****	**−0.625 ****
**Fold change in di- vs. mononucleosomal cfDNA**	**Alu-J**	**0.493 ***	0.386	−0.212	**−0.486 ***	−0.208
	**Alu-S**	0.255	0.110	−0.240	**−0.632 ****	−0.357
	**Alu-Y**	0.337	0.171	−0.314	**−0.614 ****	−0.163
	**Alu (total)**	**0.433 ***	0.248	−0.326	**−0.768 ****	−0.372

Note: * *p*-value < 0.05; ** *p*-value < 0.01. ^a^ N = 21 for this parameter, N = 24 for all the other input variables in this table.

**Table 2 ijms-26-06657-t002:** Correlation between relative abundance (CPM) of repetitive DNA subfamilies in cfDNA and parameters of clinical risk.

		NLR	Wbc (×10^3^/µL)	Rbc (×10^6^/µL)	Hgb, (g/dL)	Hct (%)	PLTs (×10^3^/µL)	Creatinine (mg/dL)	eGFR (mL/min)	Troponin (pg/mL)	Nt-Probnp (pg/mL)	CRP (mg/dL)

**CPM in mononucleosomal cfDNA**	**Alu-J**	0.123	−0.110	−0.225	−0.235	−0.227	−0.056	−0.313	0.402	0.132	−0.103	−0.079
	**Alu-S**	0.003	0.056	−0.315	−0.329	−0.311	−0.147	−0.201	0.242	0.090	−0.103	−0.004
	**Alu-Y**	−0.279	−0.003	−0.001	−0.157	−0.083	0.139	−0.418 *	0.451 *	0.083	−0.247	−0.193
	**Alu (total)**	−0.094	−0.003	−0.175	−0.255	−0.211	−0.025	−0.330	0.384	0.031	−0.141	−0.036
	**L1HS**	0.158	0.121	0.276	0.362	0.381	−0.001	−0.215	0.142	−0.316	−0.141	0.218
	**L1P**	0.012	−0.111	0.070	0.398	0.342	−0.180	−0.178	0.105	−0.249	−0.282	0.114
	**Other L1 ^**	0.070	−0.050	0.010	0.255	0.223	−0.241	−0.108	0.004	−0.283	−0.391	0.071
	**SVA**	0.171	0.128	0.029	−0.059	−0.010	−0.144	−0.252	0.346	−0.047	−0.056	0.168
	**HERV**	0.163	0.380	−0.093	−0.036	−0.065	0.097	−0.055	0.015	−0.178	−0.271	0.361
	**LTR °**	−0.044	0.193	−0.097	−0.382	−0.298	0.253	−0.213	0.309	−0.040	−0.206	−0.104
	**Other ERV_like**	0.364	0.170	−0.258	−0.151	−0.163	0.063	−0.064	0.018	−0.147	0.018	0.607 *
	**ALR**	−0.463 *	−0.169	0.091	0.141	0.095	0.141	−0.020	0.049	−0.058	−0.462	−0.100
	**6kb_BSR**	0.074	−0.074	0.270	0.400	0.318	0.031	0.107	−0.136	0.018	0.168	−0.232
	**HSAT**	0.352	−0.147	0.058	−0.042	−0.013	0.183	0.223	−0.229	0.369	0.488	−0.029
	**rRNA**	0.063	−0.185	−0.092	−0.232	−0.188	0.078	−0.204	0.094	0.085	0.121	0.246

**CPM in dinucleosomal cfDNA**	**Alu-J**	0.173	0.129	0.096	0.074	0.062	0.171	−0.492 *	0.393	−0.121	0.021	0.354
	**Alu-S**	−0.021	−0.107	0.114	0.064	0.107	−0.056	**−0.526 ****	**0.578 ****	0.076	0.026	−0.204
	**Alu-Y**	−0.088	0.019	−0.134	−0.121	−0.116	−0.210	−0.504 *	0.449 *	−0.086	−0.376	−0.229
	**Alu (total)**	0.023	0.033	0.045	0.036	0.060	−0.085	**−0.594 ****	**0.590 ****	0.038	−0.124	−0.089
	**L1HS**	−0.172	0.030	0.059	0.299	0.239	−0.035	−0.395	0.368	**−0.575 ****	−0.447	−0.011
	**L1P**	−0.103	−0.333	0.110	0.340	0.283	−0.289	0.070	0.028	−0.126	0.335	−0.389
	**Other L1 ^**	0.256	0.097	−0.068	0.127	0.074	−0.296	0.194	−0.211	0.046	−0.106	−0.071
	**SVA**	−0.270	−0.256	0.104	0.227	0.205	0.166	0.024	−0.077	−0.047	0.153	0.400
	**HERV**	0.126	0.230	−0.053	−0.042	−0.066	0.061	−0.207	0.281	0.426	−0.082	−0.107
	**LTR °**	0.060	0.203	−0.200	−0.268	−0.264	0.265	−0.246	0.180	0.014	−0.541 *	0.196
	**Other ERV_like**	0.140	0.392	0.107	−0.002	−0.013	0.107	0.050	−0.033	−0.002	−0.179	0.429
	**ALR**	−0.432 *	0.107	−0.039	0.019	−0.010	0.040	0.053	−0.016	−0.194	**−0.709 ****	0.136
	**6kb_BSR**	0.127	0.052	0.136	0.105	0.072	0.187	−0.177	0.161	0.010	−0.068	−0.057
	**HSAT**	0.261	−0.218	0.111	−0.050	0.014	−0.016	0.100	−0.081	0.357	0.438	−0.246
	**rRNA**	0.215	−0.049	−0.295	−0.369	−0.326	0.029	0.114	−0.240	0.081	0.147	0.432

Note: * *p*-value < 0.05; ** *p*-value < 0.01. ^ The category “Other L1” collects CPM of reads mapping to the L1M subfamily or to the L1 consensus sequence; ° The category “LTR” collects CPM of reads mapping to long terminal repeat (LTR) 5, LTR 7 or LTR 12. N = 24 for all parameters except troponin (N= 20), Nt-proBNP (N = 16) and CRP (N = 15). NLR: neutrophil-to-lymphocyte ratio; WBC: white blood cell; RBC: red blood cell; Hgb: hemoglobin; Hct: hematocrit; PLTs: platelets; eGFR: estimated glomerular filtration rate; Nt-proBNP: N-terminal pro-B-type natriuretic peptide; CRP: C-reactive protein.

**Table 3 ijms-26-06657-t003:** Relative abundances of Alu sequences (CPM) in cfDNA from patients with or without CKD comorbidity.

		Mononucleosomal cfDNA				Dinucleosomal cfDNA			
Repeat Class	CKD (1 = CKD)	Mean [95% C.I.]	Median (IQR)	*p*-Value ^(u)^	*p*-Value ^(a)^	Mean [95% C.I.]	Median (IQR)	*p*-Value ^(u)^	*p*-Value ^(a)^
**Alu-J**	0	5.84 × 10^4^ [(5.72–5.95) × 10^4^]	5.86 × 10^4^ (3.27 × 10^3^)	0.015	**0.046**	6.67 × 10^4^ [(6.31–7.03) × 10^4^]	6.51 × 10^4^ (8.33 × 10^3^)	0.186	0.56
	1	5.59 × 10^4^ [(5.46–5.73) × 10^4^]	5.63 × 10^4^ (3.28 × 10^3^)			6.33 × 10^4^ [(6.11–6.55) × 10^4^]	6.39 × 10^4^ (4.88 × 10^3^)		
**Alu-S**	0	2.13 × 10^5^ [(2.08–2.18) × 10^5^]	2.12 × 10^5^ (8.85 × 10^3^)	0.013	**0.039**	2.56 × 10^5^ [(2.43–2.70) × 10^5^]	2.53 × 10^5^ (2.40 × 10^4^)	0.00040	**0.0012**
	1	2.05 × 10^5^ [(2.02–2.09) × 10^5^]	2.06 × 10^5^ (9.94 × 10^3^)			2.28 × 10^5^ [(2.18–2.38) × 10^5^]	2.26 × 10^5^ (1.86 × 10^4^)		
**Alu-Y**	0	5.60 × 10^4^ [(5.45–5.74) × 10^4^]	5.59 × 10^4^ (3.30 × 10^3^)	0.022	0.065	6.80 × 10^4^ [(6.38–7.22) × 10^4^]	6.66 × 10^4^ (1.08 × 10^4^)	0.119	0.357
	1	5.40 × 10^4^ [(5.30–5.50) × 10^4^]	5.39 × 10^4^ (2.40 × 10^3^)			6.39 × 10^4^ [(6.05–6.73) × 10^4^]	6.32 × 10^4^ (8.64 × 10^3^)		
**Alu (total)**	0	3.27 × 10^5^ [(3.20–3.34) × 10^5^]	3.25 × 10^5^ (1.26 × 10^4^)	**0.0025**	**-**	3.91 × 10^5^ [(3.73–4.09) × 10^5^]	3.88 × 10^5^ (4.68 × 10^4^)	**0.0048**	-
	1	3.15 × 10^5^ [(3.11–3.20) × 10^5^]	3.15 × 10^5^ (9.22 × 10^3^)			3.55 × 10^5^ [(3.41–3.70) × 10^5^]	3.58 × 10^5^ (2.62 × 10^4^)		

Note: ^(u)^ = Mann–Whitney, *p*-value, unadjusted; ^(a)^ = Mann–Whitney, *p*-value, Bonferroni-adjusted considering the multiple testing for 3 categories (Alu-J, Alu-S, Alu-Y).

## Data Availability

Dataset available on request from the authors.

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
