# Peer review of "Repeatome Analysis of Plasma Circulating DNA in Patients with Cardiovascular Disease: Variation with Cell-Free DNA Integrity/Length and Clinical Parameters"

_ijms, 2025, doi:10.3390/ijms26146657_

Round 1
Reviewer 1 Report
Comments and Suggestions for Authors
Repeatome analysis of plasma circulating DNA in patients 2 with cardiovascular disease: variation with cell-free DNA integrity/length and clinical parameters.
1. Strengths
- Innovative application of low-pass NGS for repeatome analysis in cfDNA.
- Correlation between cfDNA repetitive content and clinical biomarkers (eGFR, CKD).
- Proper use of statistical analysis with Bonferroni correction for multiple comparisons.
- Strong rationale for the use of Alu-based quantification and subfamily-level granularity.
2. Weaknesses and Points for Improvement
- The sample size (n=24) limits the generalizability of the findings; more discussion on power and effect size would improve robustness.
- The bioinformatic methods would benefit from schematic summarization of the analytical pipeline (e.g., fragment extraction, RepBase matching).
- The authors mention but do not sufficiently explore the possible interfering role of inflammation or other comorbidities.
- The discussion of repetitive DNA overrepresentation could be clarified with more quantitative comparison to other studies.
- Clarify the thresholds used for read alignment and repeat classification (e.g., identity cutoffs, coverage filters), as these affect repeat quantification.
- Discuss how cfDNA size distribution may influence mapping efficiency for different repeat elements.
3. Recommendations for Revision
- Expand discussion of the clinical implications of Alu-S subfamily enrichment and its potential link to genomic compartmentalization.
- Include a table or figure summarizing patient characteristics and subgroup stratification by CKD or outcome.
- Discuss the limitations of low pass sequencing in more detail, particularly in relation to mapping bias against underrepresented or poorly assembled repeats.
- Consider including a post hoc power analysis to contextualize statistical strength with current sample size.
- Add a correlation matrix or visual summary of associations between repeat content and clinical biomarkers.
- Incorporate a pipeline figure to illustrate sample processing and repeatome analysis steps.
4. Final Evaluation
Overall, this manuscript provides valuable insights into the cfDNA repeatome in elderly patients with CHF and proposes a feasible and cost-effective method for clinical biomarker discovery. The methodological design is sound, and the exploratory correlations with eGFR and CKD status are compelling. Minor improvements in methodological visualization, deeper biological contextualization (especially regarding Alu subfamilies), and discussion of sequencing biases would significantly enhance clarity and impact. With these revisions, the manuscript has strong potential for publication and may serve as a foundation for more powered, clinically validated studies in cfDNA-based diagnostics.
Author Response
We sincerely thanks the Referee for the thoughtful and encouraging feedback. Please find the enclosed Word file containing our point by point response.

Reviewer 2 Report
Comments and Suggestions for Authors
Authors show an interesting study regarding the correlation between the cfDNA structure and CV risk, in a population of patients admitted due to congestive heart failure. In a detailed analysis Authors showed different distribution of major repetitive DNA classes, and correlated different Alu families with patients clinical parameters. Interestingly, Alu DNA abundance correlated with eGFR and negatively with CKD comorbidities.
Comments:
1) if possible please be more specific, e.g. L30: CKD comorbidities (which?), L76: '...abundance of repetitive cfDNA ... can be variable between... patients with different clinical conditions' (which?);
2) can you please explain all abbreviations when used for he first time, e.g.'GRCh38' in L107;
3) Table 1: data are shown differently, separated with commas and dots;
4) Table 2: please add abbreviations explanations, units to all parameters, and use 'level' or 'concentration' to all lab parameters;
5) L319: why eGFR was calculated with BIS-1 equation?
6) 4.1. section: can you please specify in which part of your study blood was collected (during admission?)/lab results represent only 1 measurement?, you've mentioned 24 months follow up;
7) Figure S3: please correct 'genoma'.
Author Response

(The authors gave the same response as above.)
